# Introducing and Implementing Genetic Assessment in Cardio-Obstetrics Clinical Practice: Clinical and Genetic Workup of Patients with Cardiomyopathy

**DOI:** 10.3390/ijms24119119

**Published:** 2023-05-23

**Authors:** Ghadeera Al Mansoori, Wael Al Mahmeed, Saleema Wani, Bashir Taha Salih, Tarek El Ansari, Fathima Farook, Zenab Farooq, Howaida Khair, Kornelia Zaręba, Nahid Al Dhahouri, Anjana Raj, Roger S. Foo, Bassam R. Ali, Fatma Al Jasmi, Nadia Akawi

**Affiliations:** 1Department of Cardiology, Sheikh Shakhbout Medical City, Abu Dhabi 11001, United Arab Emirates; gmansoori@ssmc.ae; 2Heart, Vascular & Thoracic Institute, Cleveland Clinic, Abu Dhabi 112412, United Arab Emirates; walmahmeed@seha.ae; 3Department of Obstetrics & Gynecology, Corniche Hospital, Abu Dhabi 3788, United Arab Emirates; saleemaw@seha.ae (S.W.); bashirs@seha.ae (B.T.S.); tansari@seha.ae (T.E.A.); fatimaf@seha.ae (F.F.); 4College of Medicine and Health Sciences, Khalifa University, Abu Dhabi 127788, United Arab Emirates; 100052893@ku.ac.ae; 5Department of Obstetrics & Gynecology, College of Medicine and Health Sciences, United Arab Emirates University, Al Ain 15551, United Arab Emirates; hkhair@uaeu.ac.ae (H.K.); kornelia3@poczta.onet.pl (K.Z.); 6Department of Genetics and Genomics, College of Medicine and Health Sciences, United Arab Emirates University, Al Ain 15551, United Arab Emirates; 202090177@uaeu.ac.ae (N.A.D.); anjanaajay@uaeu.ac.ae (A.R.); bassam.ali@uaeu.ac.ae (B.R.A.); aljasmif@uaeu.ac.ae (F.A.J.); 7Cardiovascular Research Institute, National University Healthcare Systems, Singapore 117599, Singapore; roger.foo@nus.edu.sg; 8Genome Institute of Singapore, Agency for Science, Technology and Research, Singapore 138672, Singapore; 9Division of Cardiovascular Medicine, University of Oxford, Oxford OX3 9DU, UK

**Keywords:** pregnancy, cardio-obstetrics clinic, preconception counselling, cardiovascular disease, inherited cardiomyopathy, genetics

## Abstract

Cardiovascular disease (CVD) during pregnancy varies significantly worldwide, influenced by factors such as access to healthcare, delayed diagnosis, causes, and risk factors. Our study sought to explore the spectrum of CVD present in pregnant women in the United Arab Emirates to better understand this population’s unique needs and challenges. Central to our study is an emphasis on the importance of implementing a multidisciplinary approach that involves the collaboration of obstetricians, cardiologists, geneticists, and other healthcare professionals to ensure that patients receive comprehensive and coordinated care. This approach can also help identify high-risk patients and implement preventive measures to reduce the occurrence of adverse maternal outcomes. Furthermore, increasing awareness among women about the risk of CVD during pregnancy and obtaining detailed family histories can help in the early identification and management of these conditions. Genetic testing and family screening can also aid in identifying inherited CVD that can be passed down through families. To illustrate the significance of such an approach, we provide a comprehensive analysis of five women’s cases from our retrospective study of 800 women. The findings from our study emphasize the importance of addressing maternal cardiac health in pregnancy and the need for targeted interventions and improvements in the existing healthcare system to reduce adverse maternal outcomes.

## 1. Introduction

Recent reports from the United States of America and Europe still refer to cardiovascular disease (CVD) as the most significant single cause of maternal deaths [1,2]. Pregnancy is associated with a complex series of cardiovascular changes necessary to support a growing fetus [3]. Although the heart rate (HR) increases across gestation, this is more evident during the third trimester of pregnancy, with almost a 15% to 20% increase compared to pregnancy pulse rate [4]. The cardiac output progressively rises to about 30% to 50%, with increases starting in the first trimester and reaching their nadir between 16 and 28 weeks [4]. During normal pregnancy, there is an increase in arterial compliance and extracellular fluid volume with a notable decrease in blood pressure (BP) and total peripheral resistance [5]. Labor and delivery are critical periods for pregnant women with CVD since uterine contractions increase venous flow and cardiac output [6]. Pain and stress of labor increase BP and HR secondary to raised sympathetic tone [7,8,9].

Cardiomyopathy, a heart muscle disease, is the most prevalent form of CVD encountered during pregnancy, particularly peripartum cardiomyopathy (PPCM) which may appear during the month before or after childbirth [10,11]. Transient cardiomyopathy is a reversible form of the disease typically caused by a specific event or condition, such as pregnancy. Still, once the underlying cause is treated or resolved, the heart muscle typically returns to normal function [12]. However, inherited cardiomyopathy, on the other hand, is a genetic condition that is passed down through families. There are several types of inherited cardiomyopathy, including hypertrophic cardiomyopathy (HCM), dilated cardiomyopathy (DCM), arrhythmogenic right ventricular cardiomyopathy (ARVC), and restrictive cardiomyopathy (RCM). DCM (Phenotypic Series OMIM#PS115200) is the most common type of inherited cardiomyopathy seen in pregnancy [13]. Characterized by dilation of the chambers and subsequent impaired contraction, DCM leads to progressive heart failure, sometimes requiring a heart transplant. Left ventricular (LV) dysfunction and/or the New York Heart Association (NYHA) functional class III or IV predict an increased risk of adverse events [10]. HCM (OMIM#PS192600), characterized by increased LV wall thickness not explained otherwise by loading conditions, is also common but not as dangerous in pregnancy [14]. Severity may vary from asymptomatic to severe, often diastolic heart failure, LV outflow tract obstruction (LVOTO), arrhythmias, or sudden cardiac death. Treatment, management, and follow-up protocols for pregnant women with presumed transient cardiac manifestations due to pregnancy-associated stress are less aggressive than for women with a strong family history of CVD and/or genetic predisposition [15,16]. Gravid women poorly tolerate the unique hemodynamic challenges of pregnancy and undergo rapid decompensation or acute cardiac events. Early diagnosis and intervention remain key [10].

Amid the notable global awareness of the significant contribution of CVD to maternal and neonatal mortalities and morbidities, there have been no studies on CVD among women in the United Arab Emirates (UAE). This study seeks to shed light on familial and inheritable patterns of clinical presentation of CVD among well-defined high-risk pregnancies in a multidisciplinary Cardio-Obstetric Clinic (COB) in the UAE. We reviewed patients exhibiting an inheritable clinical presentation pattern of CVD diagnosed during pregnancy or preconception counseling at the COB at Corniche Hospital, one of the largest maternity tertiary centers in the Middle East. Selected examples of high-risk pregnancy cases with cardiovascular complications were evaluated and discussed in detail. Our results highlight the importance of obtaining detailed family histories and implementing an advanced genetic workup in COB practice to identify high-risk women with underlying inheritable CVD.

## 2. Results

### 2.1. Overview of the Corniche Hospital Cardio-Obstetrics Clinic Referrals

From 800 new referrals to the COB in the past decade, 206 (25.8%) had significant cardiac conditions, including congenital heart disease (CHD; 26.2%), valvular heart disease (VHD; 48.2%), prosthetic heart valve (PHD; 6.8%), cardiomyopathy (CM; 13.17%), ischemic heart disease (IHD; 26.2%), and pulmonary hypertension (PHT; 0.5%). Among 226 pregnancy outcomes in the 206 patients, we documented 10 (4.42%) miscarriages, 10 (4.42%) terminations, and 10 (4.42%) defaulted follow-ups. The 206 delivered 211 babies (3 sets of twins and one set of triplets). In this cohort, we encountered 23.1% maternal cardiac events including heart failure, arrhythmia, and death, with a significantly high occurrence rate in the WHO class II (10.7%) and III (41%) groups [17]. In this cohort, the cesarean (C-sections) rate was 41.3%, prematurity was reported to be 16.8%, and intrauterine growth restriction (IUGR) was 8.2%. Approximately 17% of all C-sections were for pure cardiac indications; the rest were due to multifactorial reasons. Out of the 27 patients in the CM group, 12 patients were diagnosed with PPCM, 6 with HCM, 5 with DCM, and 4 with CM as a secondary manifestation of myocarditis or hypertension. The CM group had the highest prematurity (29%) and C-section rate (58.1%). No maternal or fetal mortalities occurred. Among this group, there were 47 (22.3%) Neonatal Intensive Care Unit admissions, primarily due to prematurity. Seven babies had a congenital heart defect, and one neonatal death was documented due to hypoplastic left heart syndrome.

From 800 referrals, we selected 5 patients diagnosed with cardiomyopathy during pregnancy and 1 patient identified during preconception counseling at the clinic (Table 1). We conducted a complete clinical evaluation and genetic testing for all five patients and available family members.

### 2.2. Clinical and Genetic Workups of Five Patients

#### 2.2.1. Patient-1

Clinical Assessment: Patient-1 is a 32-year-old female with five previous pregnancies. During her first three pregnancies, she had mild thrombocytopenia without heart disease. In her fourth pregnancy, she presented with cardiac symptoms at 35 weeks gestation and was diagnosed with peripartum DCM. An elective C-section was delivered to her at 36 weeks. During her fifth pregnancy, she was referred to the COB. She experienced palpitations without signs of heart failure throughout the pregnancy and was delivered vaginally at 37 weeks in May 2021.

Family History: The patient (III:6; Figure 1A) is from an extended Emirati family with distantly related parents (from the same tribe). The patient has a strong family history suggestive of CM. One sister (Patient-2, III:4, Figure 1A), an aunt (II:12), and two uncles (II:7, II:8) are confirmed cases of CM, with one uncle having a sudden cardiac death (II:7). Her brother (III:5) also died suddenly at age 25-years-old. Her father and uncle (II:6, II:9) have coronary artery disease. Her mother (II:5) was killed in a motor vehicle accident at age 50 without a documented medical history showing a similar phenotype.

Cardiac Diagnostic Investigations: Transthoracic echocardiography (echo) during her pregnancy showed mildly reduced global systolic function with an ejection fraction (EF) of 40% with trace mitral and tricuspid regurgitation. Electrocardiogram (ECG) showed sinus tachycardia with PR 118 ms, QRS 100 ms, and QTc 414 ms. Her BP was reportedly 130/78 mmHg, and her NT-proBNP level was 92.4 ng/L (<400 ng/L Normal level).

Clinical Genetic Testing: Untargeted screening for the disease-causing genetic variant(s) was done using a whole-exome sequencing assay for Patient-1 (III:6, Figure 1A,B). One homozygous variant was found to be a non-benign, rare (MAF < 1%) variant falling in inherited CM-causing genes. This variant is causing an in-frame deletion of 3 nucleotides in the *NEXN* gene ((NM_144573.4:c.1582_1584del(p.Glu528del)). This homozygous variant was also detected by exome and Sanger sequencing in the patient’s sister (Patient-2, III:4; Figure 1A,B), a confirmed case of DCM. This homozygous deletion was missing from our cohort of 343 in-house exomes, including the exomes of 33 patients with documented CM and their family members. It is also not listed in the homozygous state in gnomAD v2.1.1, spanning 125,748 exome sequences and 15,708 whole-genome sequences from unrelated individuals. The position is conserved (phyloP = 9.77, Gerp++ = conserved, CADD =22.5), and MetaDome predicts amino acid position 435 and onwards to be intolerant to changes (Figure 1C) [18]. GLU528 falls in a repetitive sequence of glutamic acid residues starting from 525 to 528 that form several hydrogen bonds with surrounding amino acids, possibly indicating a role in protein folding (Figure 1D). Other heterozygous pathogenic variants and variants of unknown significance falling within known inherited CM genes that might contribute to this phenotype are also shown in Table 1.

Clinical Management/Follow-up/Treatment: Patient-1 continues to be followed up at a heart failure clinic with the most recent echo showing EF of 30–35% and ECG with 82 bpm, PR 132, QRS 97, QT/QTc 391/429, sinus rhythm. The patient is currently on targeted heart failure therapy and a comprehensive family screening was offered for her offspring. She is under evaluation to assess her need for Implantable Cardioverter-Defibrillator (ICD).

#### 2.2.2. Patient-2

Clinical Assessment: Patient-2 is a 38-year-old sister of Patient-1. She had three pregnancies with spontaneous vaginal deliveries. The third delivery, however, was complicated with DCM on day eight postpartum in 2009. The fourth pregnancy was in 2020. She was seen at the high-risk COB at 11 weeks of gestation. The LVEF was ~25–28%. The patient was offered therapeutic termination of the pregnancy as it carried significant morbidity and mortality to the patient, but she declined and preferred to continue with the pregnancy. During the peak hemodynamic changes of pregnancy, the patient experienced prolonged episodes of dyspnea, orthopnea, and inability to sleep and had an emergency C-section at 32 weeks for fetal IUGR and maternal decompensated heart failure. The LVEF was around 25%.

Family History: Patient-2 (III:4; Figure 1A) has a positive family history. As indicated earlier, she is the sister of Patient-1 (III:6; Figure 1A), who was also diagnosed with DCM during pregnancy. A comprehensive family screening revealed that her husband (III:3) is her first-degree cousin. Her husband’s brother (III:2) and his daughter (IV:1) were also confirmed DCM cases.

Cardiac Diagnostic Investigations: Echo at the early onset of pregnancy at 7 weeks revealed a mildly dilated left atrium and ventricle with moderate systolic dysfunction and EF of 35%. Mild mitral regurgitation secondary to CM was also observed. Subsequent echo assessment demonstrated deteriorating left ventricular systolic function reaching less than 25% prior to delivery (around 32 weeks). ECG showed sinus rhythm with a prolonged PR interval of 211 ms, QRS 109 ms, and QTc 438 ms. Her BP was reported to be 120/72 mmHg, and her NT-proBNP level was 33.7 ng/L which then peaked at 153 ng/L.

Clinical Genetic Testing: Whole-exome sequencing was performed for Patient-2, revealing a homozygous deletion of three nucleotides in the *NEXN* gene ((NM_144573.4:c.1582_1584del(p.Glu528del)) shared by Patient-1. The same variant was found in the four tested children of Patient-2 (IV:2-5) but in a heterozygous state (Figure 1A,B). Other pathogenic variants and variants of unknown significance falling within heritable CVD genes that might contribute to the patient phenotype are also shown in Table 1.

Clinical Management/Follow-up/Treatment: The patient’s postpartum echo revealed a severely decreased LVEF of less than 25% with severe LV systolic dysfunction and mild to moderate mitral regurgitation. The patient was resumed on targeted heart failure therapy and continues to be followed up at the heart failure clinic. Her four offspring underwent comprehensive clinical and genetic workup.

#### 2.2.3. Patient-3

Clinical Assessment: Patient-3 (II:2; Figure 2A) is a 33-year-old female referred as a high-risk pregnancy. She was diagnosed with obstructive HCM in 2018 and started on beta blockers. Her first pregnancy was unremarkable before discovering the heart condition, but her second pregnancy was complicated with reported palpitations. A 24-Hour Holter Monitoring showed basic sinus rhythm with the minimum heart rate reported as 43 bpm and maximum as 150 bpm. Occasional, isolated premature ventricular contractions (PACs), multifocal atrial contractions (PVCs), and ventricular couplets were noted, along with the low atrial ectopic rhythm. The patient had a vaginal delivery assisted by forceps at 40 weeks and was admitted to the High Dependency Unit for observation.

Family History: The father (I:1; Figure 2A) of Patient-3 was diagnosed with CM at the age of 30. No history of sudden cardiac death.

Cardiac Diagnostic Investigations: Transthoracic echo at pregnancy revealed an EF of 55%, IVS thickness of 3.2 cm, systolic anterior motion (SAM)/LVOT obstruction, and a peak gradient of 60–90 mmHg, with a normal pulmonary artery (PA) pressure. ECG showed sinus rhythm with a PR interval of 153 ms, QRS 92 ms, and QTc 473 ms. BP was reported to be 112/65 mmHg, and NT-proBNP level was 468 ng/L.

Clinical Genetic Testing: Whole-exome sequencing was performed for Patient-3, her husband, and two sons. A heterozygous pathogenic variant was detected in the *MYH7* gene in the patient (II:2; Figure 1A) as well as one of her young asymptomatic sons (III:1; Figure 1A). In silico analysis supports that this missense variant has a deleterious effect on protein structure/function In silico analysis supports that this nonsynonymous variant in *MYH7* (NM_000257.4:c.4066G>A (p.Glu1356Lys)) has a deleterious effect on protein structure/function according to ACMG classification criteria as shown in Table 1. This variant was confirmed in this family by Sanger sequencing (Figure 2B) and was not found in gnomAD or our in-house cohort of exomes.

Clinical Management /Follow-up/Treatment: The patient’s postpartum echo revealed an interventricular septal end diastole and end systole thickness of 3.4 cm with a peak gradient of 121 mmHg. A cardiac magnetic resonance imaging (MRI) showed average LV volumes with supra-normal EF of 79% with SAM of the mitral valve and LVOT obstruction at rest. Asymmetric septal hypertrophy affecting the basal to mid-anterior wall and septum (max 30mm) was noted, with patchy myocardial scarring in the hypertrophied segments. Risk stratification for sudden cardiac death was done for the patient, and consideration for ICD was discussed, but due to lack of non-sustained VT or any arrhythmic syncope in the past and with a maximum septal thickness of <30 mm, the indication is low. The patient continues to be followed up regularly at the cardio-genetic clinic for regular evaluations. Her offspring received clinical and genetic screening and are enrolled at the COB for future cardiac assessment.

#### 2.2.4. Patient-4

Clinical Assessment: Patient-4 (IV:2; Figure 3A) is a 25-year-old female who presented at 6 months’ gestation with recurrent exertional palpitations and shortness of breath. Echo revealed low-normal left ventricular systolic function with an estimated EF of 50% with no wall motion abnormalities and no significant valve lesions. In response to her symptoms of palpitations, a Holter was completed which revealed sinus rhythm with an average HR of 89 bpm, a maximum HR of 130 bpm, and a minimum of 60 bpm. 134 premature ventricular contractions representing 0.1 of the total beat count. There were three runs of wide complex rhythm suggestive of non-sustained ventricular tachycardia, and the longest was a five-beat duration of 182 bpm. The patient was commenced on beta blockers.

At 34 weeks, the patient presented to the emergency department with symptoms of decompensated heart failure, and her echo revealed severe left ventricular dysfunction with EF of 18–20%, with global hypokinesia and mildly dilated left atrium, mild-moderate mitral regurgitation, moderately reduced right ventricular dysfunction. She was delivered via vacuum-assisted delivery at 34 weeks. Subsequently, she continued to be in biventricular systolic heart failure status and was followed up by a heart failure clinic.

Family History: The patient has a positive family history of DCM, where two of her brothers with DCM (IV:5, IV:6; Figure 3A) died at ages 11 and 14, secondary to heart failure. Another brother (IV:4; Figure 3A) had a cardiac transplant for worsening heart function at the age of 25 years. Several sisters of the patient have some degree of cardiomyopathy. The parents (III:1, III:2; Figure 3A) are first cousins, and both have DCM. Two of the patient’s maternal uncles also died suddenly in their 30s (III:3, III:4; Figure 3A).

Cardiac Diagnostic Investigations: Transthoracic echo at early pregnancy revealed an EF of 50% with no significant valve lesions or wall motion abnormalities. ECG showed sinus tachycardia with a PR interval of 129 ms, QRS 80 ms, and QTc 421 ms. Her blood pressure was reportedly 120/72 mmHg, and her NT-proBNP level was 1896 ng/L. Subsequent Holter revealed sinus rhythm with an average HR of 89 bpm, a maximum HR of 130 bpm, and a minimum of 60 bpm. 134 premature ventricular contractions representing 0.1 of the total beat count. There were three runs of wide complex rhythm suggestive of non-sustained ventricular tachycardia, the longest was a five-beats duration of 182 bpm. Evidence of deteriorating bi-ventricular systolic function with EF around 18–20% at 34 weeks gestation. Her NT-proBNP also peaked at 4111 ng/L at 34 weeks and trended down to 1082 ng/L postpartum.

Clinical Genetic Testing: One of the patient’s brothers (IV:12; Figure 3A) and both parents (III:1, III:2; Figure 3A) underwent whole-genome sequencing in a certified clinical lab (Centogene, Cambridge, MA, USA). The only variant that was found to be the plausible cause of IV:12 DCM phenotype was a homozygous nonsynonymous substitution in *MYPN* gene ((NM_001256267.1):c.65C>G (p.Ala22Gly)) (Table 1). The variant was co-segregated with the DCM (Figure 1A,B). Both parents carried the heterozygous form of this variant. Target Sanger sequencing of available family members revealed that Patient-4 (IV:2) and some of her other siblings were carriers of the heterozygous variant (IV:4, IV:7, IV:8) (Figure 1A,B). This variant is rare (MAF = 0.000167) and was predicted to be pathogenic by nine computational predictions tools, including DANN, EIGEN, FATHMM-MKL, LIST-S2, M-CAP, MVP, MutationAssessor, MutationTaster, and SIFT VS. Two benign predictions from BayesDel_addAF and PrimateAI were reported. Ala22 forms multiple hydrogen bonds with other amino acids within the MYPN protein that could be disrupted by its deletion (Figure 3C).

Clinical Management/Follow-up/Treatment: The patient’s postpartum course was complicated by decompensated heart failure one month after delivery. The echo showed severely reduced LV systolic function with an EF of 20%, mild mitral regurgitation, and moderately reduced RV systolic function. The patient continues heart failure therapy with regular follow-ups at the heart failure clinic and is awaiting a heart transplant.

#### 2.2.5. Patient-5

Clinical Assessment: 18-year-old female (II:1; Figure 4A) with a known history of severe mitral valve insufficiency and DCM presented for preconception counseling. The patient had been on complete heart failure therapy since childhood.

Family History: The patient’s younger sister (II:2, Figure 4A) died suddenly in the context of DCM.

Cardiac Diagnostic Investigations: Transthoracic echo revealed severely dilated LV with mild global systolic dysfunction, LVEF of 40%, and a thickened mitral valve with mild mitral regurgitation. Her BP was reportedly 120/75 mmHg, and her NT-proBNP level was 15.9 ng/L. ECG showed sinus rhythm with repolarization abnormalities, global ischemia, 95 bpm, PR 161 ms, QRS 90 ms, QTc 335 ms. Exercise stress test using the Bruce protocol demonstrated good functional capacity with no ST changes, arrhythmias, or symptoms.

Clinical Genetic Testing: Exome sequencing of Patient-5 revealed a heterozygous variant in *RBM20* (NM_001134363.3):c.19A>G (p.Met7Val). This variant was not detected in our in-house exomes or any subpopulation of the gnomAD. It was reported once in ClinVar as a variant of unknown significance and 5 in-silico tools predicted this variant as damaging (SIFT, FATHMM, FATHMM-MKL, M-CAP, PrimateAI). Yet, Patient-5 inherited the nonsynonymous variant from her mother (I:2; Figure 1B). Her father carried the wild-type genotype (I:1; Figure 1C). No other family members were available for sequencing. Different rare variants falling in CVD genes that are not present in our in-house exomes are listed in Table 1.

Clinical Management/Follow-up/Treatment: The patient continues to be followed up regularly at the cardiology clinic for regular evaluations. Clinical screening and genetic testing for her siblings were advised.

## 3. Discussion

Cardiac conditions in pregnant women vary significantly worldwide, with differences in forms and severity attributed to factors such as access to healthcare, delayed diagnosis, and the prevalence of risk factors [19]. In this study, we examined the spectrum of cardiac conditions present in pregnant women in the UAE. Additionally, we presented a detailed description of five women who were retrospectively selected to underscore the significance of a multidisciplinary approach in the diagnostic workflow at the COB. This approach is essential in minimizing adverse cardiac outcomes and related mortality during pregnancy.

### 3.1. Spectrum and Impact of Cardiovascular Disease Seen in Women from the United Arab Emirates

The ROPAC study, which evaluated 2742 women with cardiac conditions from 39 countries in Europe and America, found that CHD (58.2%), VHD (31.4%), CM (5.9%), IHD (0.6%), aortopathy (3.6%), and PHT (0.3%) were the most prevalent conditions [20]. In our cohort of 206 women with cardiac diseases in the UAE, VHD (48.2%) was the most prevalent condition, followed by CHD (26.2%). This difference can be attributed to the recent establishment of a defined service for CHD in specialized COB and the inclusion of women from countries where rheumatic heart disease is still prevalent. Conversely, IHD was more common in women in the UAE (26.2%), likely due to the higher prevalence of metabolic syndrome and other risk factors. Notably, a significantly increased occurrence of adverse maternal outcomes, such as heart failure, was observed in WHO class II (10.7%) and III (41%) compared to the ROPAC study findings. This difference could also be linked to patients’ lifestyle and risk factors.

While the ROPAC study compared emerging countries to developing countries, it is essential to note that our results focus on a specific country, the UAE, and cannot be generalized to other countries. According to the ROPAC study, the occurrence of cardiac events in emerging and developing countries was estimated to be 12.8% and 36.5%, respectively. Our study found that the incidence of maternal cardiac events among women in the UAE was 23.1%. These findings suggest the need for specific improvements to the existing healthcare system for managing pregnancy-related CVD. These improvements may include increasing women’s awareness of CVD risk in pregnancy, obtaining detailed family histories of patients, implementing genetic testing and family screening, and introducing a multidisciplinary approach to managing patients with cardiac conditions during pregnancy [21].

Premature infants face long-term deleterious health outcomes, as demonstrated over the past decade [22]. Our cohort of women with CVD showed an increased rate of preterm delivery, defined as less than 37 weeks of gestation. This finding was particularly evident in patients with cardiomyopathy, which has been associated with adverse maternal and fetal outcomes [23,24]. The ESC guidelines recommend vaginal delivery in women with heart disease is vaginal, as C-section delivery is related to maternal and fetal short- and long-term side effects [25,26]. However, we and other studies have recorded high rates of C-section delivery among women with cardiac conditions, especially those with cardiomyopathy [21,24,27].

### 3.2. Clinical and Genetic Risk Assessment and Care during Pregnancy at the COB

Heart failure and arrhythmia are the most common complications of CM during pregnancy [12]. Pregnancy involves complex cardiovascular changes necessary to support fetal growth [3]. In the five women presented in this study, these changes tipped the balance of their cardiac compensatory mechanisms. They made them vulnerable to significant cardiac morbidities resulting in severe long-term complications and chronic congestive heart failure. It became apparent that they could not cope with the physiological adaptions of pregnancy, as they began experiencing symptoms from the late second trimester of pregnancy or immediately postpartum. Furthermore, they had long-term significant cardiac morbidities. Our patients displayed similar complications. Patients 1 and 2 had systolic heart failure with reduced ejection fraction, while Patients 3 and 4 had abnormal cardiac rhythm consistent with non-sustained ventricular tachycardia.

Idiopathic PPCM is the most common CM seen during pregnancy, resembling familial DCM [28]. The prognosis of PPCM depends on the return of left ventricular size and function to normal within six months after delivery [29]. It is estimated that about 50% of patients with PPCM recover without complications. Yet, our patients fall within the 50% who failed to recover their systolic function post-delivery. Therefore, one of the critical components of managing cardiovascular concerns in pregnancy is understanding if the associated cardiovascular complications are inherited or acquired via obtaining a detailed family history and performing genetic testing. The genetic basis of familial CM is complex. It involves multiple genes with varying functions and has been complicated by its heterogeneous etiologies, reduced penetrance, and variable expressivity within the same family members. Differences in disease expression must arise from patient-specific other genetic variations (modifiers) or stimuli (e.g., pregnancy).

With advances in next-generation sequencing technologies, the genetic architecture of CM is expanding rapidly, leading to the identification of hundreds of genes implicated in the pathogenesis, with substantial overlap among them [30,31]. These genes encode proteins in various cardiac myocyte compartments performing diverse functions related to contraction, structural integrity, ion channels, and molecular chaperones and include proteins found in the nuclear membrane, sarcomere, Z-disk, desmosome, and cytoskeleton. In our study, sequencing the exomes and the genomes of the five women revealed several possible CM-predisposing genetic variants in disease-related genes, including *NEXN*, *MYH7*, *MYP*, *RBM20,* and others listed in Table 1. These genes encode proteins in various cardiac myocyte compartments, and their defects can cause different types of CM. Most reported disease-related variants in these genes were heterozygous such as the variant detected in *MYPN* and *MYH7*, while some were biallelic [32,33]. The nonsynonymous variant detected in *MYH7* in Patient 3 was predicted to be pathogenic by all in silico algorithms and was confirmed in other CM patients [34]. In-vitro and in-vivo functional studies suggested an impact of this variant on myosin structure but not contractibility [35]. The evidence from the in-silico analysis of the *NEXN* variant that was detected in Patients 1 and 2 was conflicting, the variant was not reported in a homozygous state in healthy individuals, and it has been described in two infants with severe DCM and one Caucasian pediatric patient with HCM [36,37,38,39]. Another interesting candidate in Patients 1 and 2 is a heterozygous variant in *ABCC6* (p.Arg1164Gln), a gene known to cause multisystemic disease affecting tissues rich in elastic fibers as the cardiovascular system called Pseudoxanthoma Elasticum (PXE) [40,41]. The identified variant is reported in PXE patients but in homozygous or compound heterozygous states with another *ABCC6* variant [40]. In our patients, the second hit was not detected, and the key clinical features of PXE were missing; however, this variant could still contribute to their cardiac manifestations and patients require full surveillance by a rheumatologist for further counseling.

Conversely, the variants detected in the *MYPN* and *RBM20* genes are not very strong, and their pathogenicity still needs to be confirmed. However, next-generation sequencing in both Patients 4 and 5 revealed no other strong candidates. The *MYPN* variant co-segregated with disease in affected family members and in-silico analysis revealed contradicting results. Other variants in CVD-related genes detected in these patients, listed in Table 1, might be modifiers.

### 3.3. Preconception Risk Assessment and Care

Preconception counseling is crucial in women with cardiac disease to ensure optimal pregnancy outcomes. The European Society of Cardiology (ESC) guidelines recommend preconception counseling to identify and manage potential risk factors that may impact pregnancy and to optimize health outcomes. However, there is still room for improvement in delivering preconception care, as highlighted in recent studies [42,43,44]. At the cardio-obstetric clinic, preconception counseling is a vital component of care for women with cardiac conditions. Exercise stress tests are one tool that can be used to evaluate a woman’s tolerance to pregnancy and labor by assessing her heart’s response to the hemodynamic challenges of pregnancy [45]. In addition, alternative methods of conception, such as preimplantation genetic testing (PGT), can be explored to reduce the risk of heritable cardiogenetic pathologies. By identifying potential risks and implementing appropriate management strategies, preconception counseling can help optimize health outcomes for women with cardiac disease.

Patient-5 was identified during preconception counseling at the COB before pregnancy in this study. An exercise stress test was performed, which showed a negative result, indicating a low risk for adverse cardiac events. Options regarding contraception and alternative methods of conception were explored. PGT, which involves screening embryos for pathogenic variants before implantation during in-vitro fertilization, was recommended as the most beneficial option for disease prevention of heritable cardiogenic pathologies based on the patient’s genetic testing results [46].

### 3.4. Family Clinical Screening and Genetic Risk Assessment

Our study highlights the importance of screening families of women with CVD for at-risk relatives to identify inherited cardiac abnormalities before the onset of clinical symptoms [47]. It also shows that genetic testing for at-risk relatives can be valuable in improving counseling and identifying targeted testing options, even before implantation, if the family variant is pathogenic. The patients in our study had an undisclosed positive family history. Some likely causative variants detected in patients were carried by their asymptomatic offspring following a Mendelian inheritance pattern, such as *MYH7* pathogenic variant in Patient-3. By performing thorough family screening and using three-generation pedigrees to appreciate the inheritance pattern (limitations due to de novo and recessive variants, reduced penetrance, and variable expressivity can hinder interpretation [47,48]) and incorporating targeted genetic testing for at-risk relatives, clinical outcomes can be improved.

### 3.5. Strengths and Limitations of the Study

This study represents the first exploration of the spectrum and complications of CVD in pregnant women in the UAE. It is also the first investigation of CM’s genetics and clinical characteristics in this country, particularly concerning pregnancy, preconception counseling, and family screening. The study’s limitations include a small sample size, a single-center experience, and the recent introduction of a defined service for CHD in the specialized COB. Additionally, there are known limitations associated with high-throughput genetic testing, such as the complex inheritance of CVD, large phenotypic variability, variable age of onset, and severity. Variant’s classification is still a work in progress, making interpretation challenging. The absence of functional validation further complicates the interpretation of variants, as it may be difficult to determine whether a variant significantly affects protein function. Further clinical and molecular studies are necessary to improve our understanding of CVD in the country and shape the current healthcare workflow accordingly.

## 4. Materials and Methods

### 4.1. Material

This study was performed in the Corniche Hospital, a leading tertiary women and newborn hospital in the UAE with an annual delivery rate of 7800. We reviewed pregnancy outcomes in patients with heart disease attending the COB since it was established in 2007 until December 2022. We selected four patients diagnosed with cardiomyopathy during pregnancy and one patient identified during preconception counseling at the clinic. We conducted a complete clinical evaluation and genetic testing for all five patients and available family members.

The study followed the principles of the Declaration of Helsinki and was approved by the Institutional Ethics Committee of the Abu Dhabi Department of Health (ethical approval number DOH/CVDC/2022/82). Informed consent was obtained from all subjects involved in the study. Patients were diagnosed according to the criteria of the World Health Organization (WHO, Geneva, Switzerland) and the European Society of Cardiology (ESC) guidelines [3,48].

### 4.2. DNA Isolation and Whole-Exome Sequencing

Whole blood samples were obtained from patients in EDTA tubes. DNA was extracted from blood using the Qiacube Automated DNA Extraction system with the Qiamp DNA Mini kit (Qiagen, Hilden, Germany). Genomic DNA was fragmented by LE-220 plus Focused Ultrasonicator (Covaris, Woburn, MA, USA). The libraries were prepared using the TruSeq Exome kit (Illumina, San Diego, CA, USA). The sequencing was performed on genomic DNA using NovaSeq 6000 system according to the manufacturer’s instructions (Illumina, San Diego, CA, USA). In-house developed bioinformatics tools and utilities were applied, including bioinformatics pipeline, base calling, primary filtering of low-quality reads and probable artifacts, and annotation of variants.

### 4.3. Variant Identification, Filtering, and Classification

Variant tertiary analysis was performed in VarSeq software 2.2.3 (Golden Helix, Bozeman, MT, USA). The following databases and in-silico algorithms were integrated and used to annotate and evaluate the impact of the variant in the context of human disease: gnomAD, ClinVar, HGMD, OMIM, dbSNP, NCIB RefSeq Genes, ExAC Gene Constraints, SIFT, PolyPhen2, PhyloP, GERP++, GeneSplicer, MaxEntScan, NNSplice. The variants were interpreted according to the ACMG guidelines and patient phenotype. The WES analysis considered all disease-causing variants in ClinVar (http://ncbi.nlm.nih.gov/clinvar; accessed on 8 May 2023), Human Genome Mutation Database (HGMD; http://hgmd.cg.ac.uk; accessed on 1 December 2022)), as well as all variants with minor allele frequency (MAF) of less than 1% in the Genome Aggregation Database (gnomAD; https://gnomad.broadinstitute.org; accessed on 8 May 2023)), in the coding regions and exon/intron boundaries ± 20 bp in the target genes. VarSome (https://varsome.com; accessed on 8 May 2023) was used for the comprehensive interpretation of the detected variants. Candidate variants falling within CVD-related genes were assessed in 343 in-house exomes, including the exomes of 33 patients with confirmed cardiomyopathy and their family members. Pathogenicity analysis of genetic variants through the aggregation of homologous human protein domains was done using MetaDome [18]. The impact of some variants was evaluated on the hydrogen bond formation of their corresponding proteins’ three-dimensional structure retrieved from UniProt.

### 4.4. Segregation Analysis

To confirm and analyze the segregation pattern of the most likely causative candidate variants for each studied patient, genomic DNA from the patient and available family members were used to amplify the region harboring the variant using standard PCR conditions followed by Sanger sequencing performed in both the forward and the reverse directions. Primers used for PCR and sequencing were designed by Primer3 (https://primer3.ut.ee; accessed on 8 May 2023). DNA sequencing was carried out with automated fluorescent sequencing on the ABI 3130xl genetic analyzer (Thermo Fisher, Waltham, MA, USA). Sequencing data were analyzed using Multiple Sequence Alignment tools (www.ebi.ac.uk, accessed on 8 May 2023).

## 5. Conclusions

The increasing complexity of cardiac abnormalities that arise during pregnancy necessitates a multidisciplinary approach to differentiate between inherited and acquired conditions, with a particular emphasis on genetic screening tests to identify heritable causes of heart disease and improve maternal and fetal outcomes. Collaboration between cardiologists, obstetricians, genetic specialists, obstetric anesthesiologists, and neonatologists is essential to optimize the patient’s cardiac status and formulate a management plan to ensure the best outcomes for both mother and the fetus. Given the high-risk nature of inherited cardiac conditions, including CM, women with such conditions require tailored, responsive care that extends across their reproductive lifespan. Therefore, the following issues should be considered in women with cardiomyopathy before pregnancy:Risk stratification of disease severity;Full understanding of the underlying etiology (inheritable, post-partum cardiomyopathy, valve-related, congenital heart disease);Risk of transmission to offspring;Projected effect of pregnancy on disease progression;Impact of medication/treatment on fetal growth during the breastfeeding period;Facilitating access to a multi-disciplinary team to prove cardio obstetric care.

## Figures and Tables

**Figure 1 ijms-24-09119-f001:**
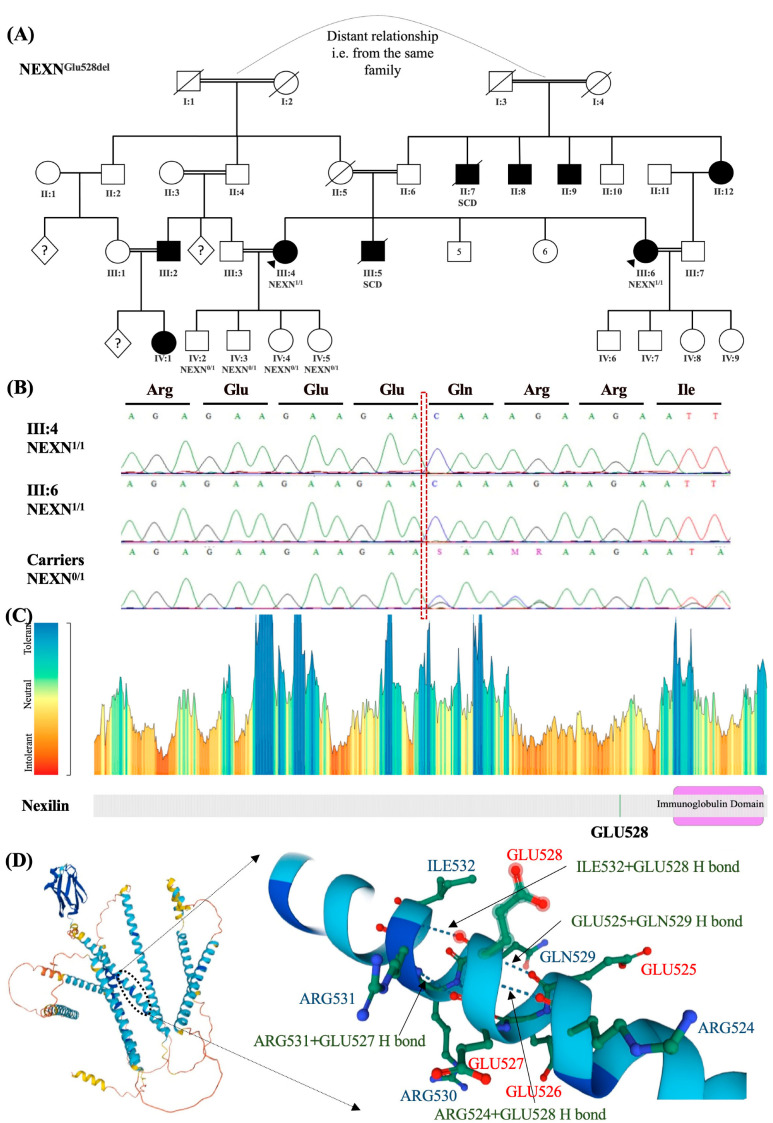
Patient-1 and Patient-2’s family pedigrees and genetic analysis results. Family pedigree showing the family history of Patient-1 (III:6) and Patient-2 (III:4) and segregation analysis results in available family members for the variant in *NEXN* gene ((NM_144573.4:c.1582_1584del(p.Glu528del)) (**A**). Circles indicate female family members; squares, male family members; diamonds, unknown gender; filled symbols, presence of the cardiomyopathy phenotype; open symbols, asymptomatic family members; arrows, the proband; symbols with a slash mark, those who have died; question marks, unknown number; 0/1 indicates heterozygosity; 1/1 indicates homozygosity. Sanger sequencing representative chromatograms showing the variant in homozygous states in the patients and a heterozygous state in one of the carriers (**B**). The c.1582_1584del at the position indicated by a red dashed box causes a homozygous deletion of a Glu residue in the patients. MetaDome web server results for the gene NEXN predicts amino acid position 435 and onwards to be intolerant to changes (**C**). The schematic diagram depicts the 3D structure of the human NEXN protein (Accession# Q0ZGT2) with a focus on the variant position. It showcases the surrounding interactions (**D**) and highlights the specific location of each amino acid within that position. The green dashed lines indicate the hydrogen bonds between the included residues. The detected variant is indicated to potentially affect Glu residues, which are highlighted in red font. Glu, Glutamate; Arg, Arginine; Gln, Glutamine; Ile, Isoleucine; H bond, Hydrogen bond.

**Figure 2 ijms-24-09119-f002:**
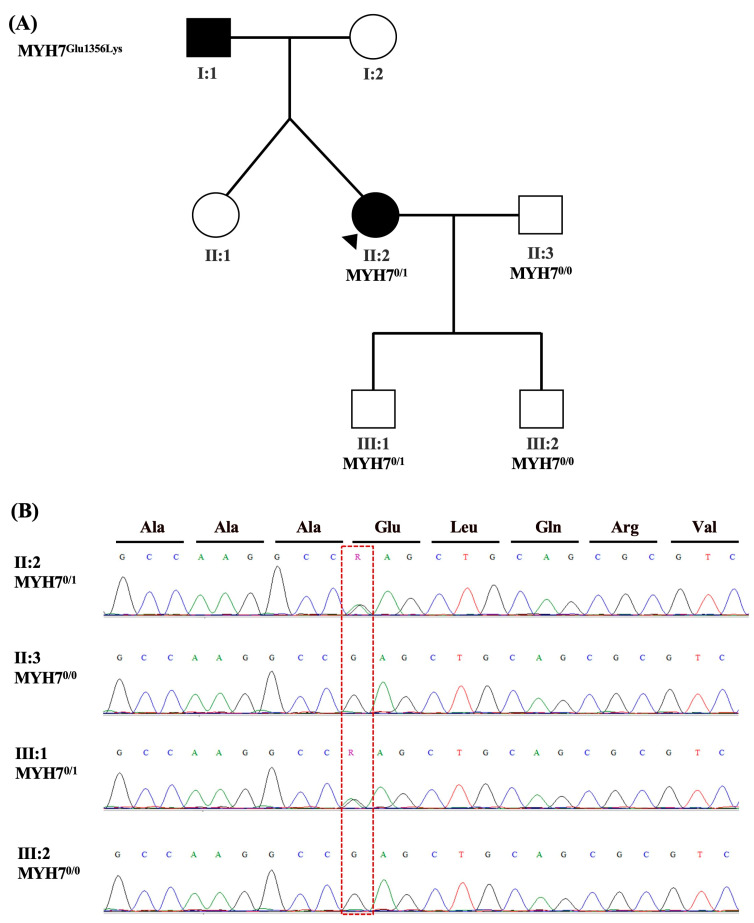
Patient-3’s family pedigree and genetic analysis results. Family pedigree showing the family history of Patient-3 (II:2) and segregation analysis results in available family members for the variant in *MYH7*(NM_000257.4:c.4066G>A (p.Glu1356Lys)) (**A**). Circles indicate female family members; squares male family members; filled symbols the presence of the cardiomyopathy phenotype; open symbols asymptomatic family members; arrows the proband; 0/1 indicates heterozygosity; and 1/1 indicates homozygosity. Sanger sequencing chromatograms confirm the presence of the variant c.4066G>A in the patient (II:2) and in one of her sons (III:1) that alters GAG (Glu) at that position to AAG (Lys) (**B**). It was absent in her spouse (II:3) and her other son (III:2). The corresponding encoded amino acid is shown above the chromatograms, and the mutated nucleotide is enclosed in a red dashed box. Ala, Alanine; Glu, Glutamate; Lys, Lysine; Leu, Leucine; Gln, Glutamine; Arg, Arginine; Val, Valine.

**Figure 3 ijms-24-09119-f003:**
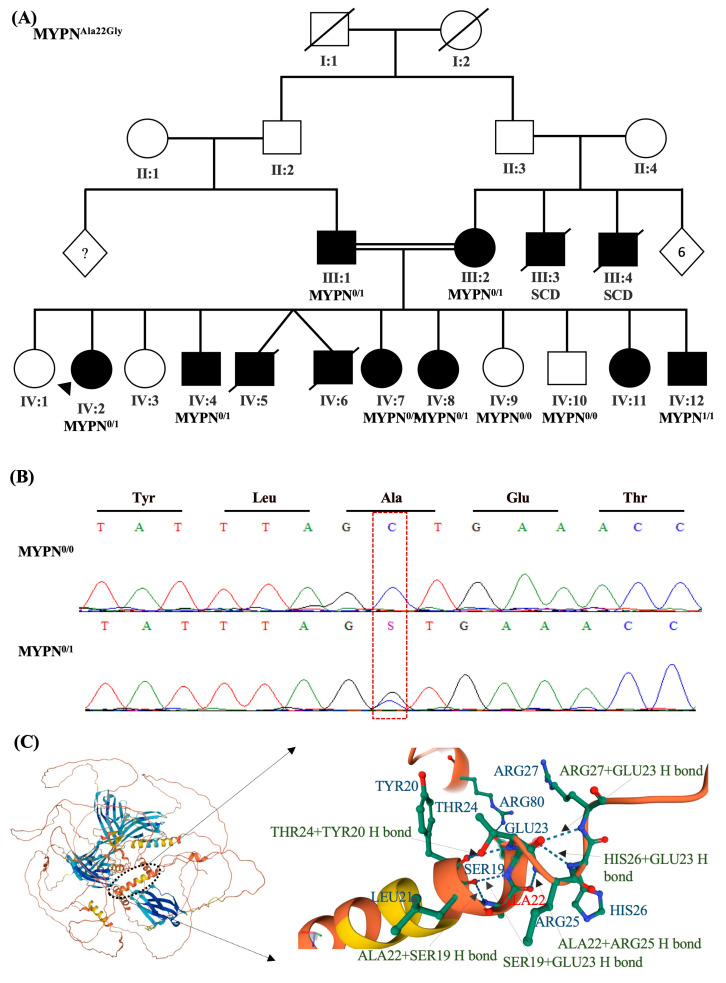
Patient-4’s family pedigree and genetic analysis results. Family pedigree showing the family history of Patient-4 (IV:2) and segregation analysis results in available family members for the variant in the *MYPN* gene ((NM_001256267.1):c.65C>G (p.Ala22Gly)) (**A**). Circles indicate female family members; squares, male family members; diamonds, unknown gender; filled symbols, presence of the cardiomyopathy phenotype; open symbols, asymptomatic family members; arrows, the proband; symbols with a slash mark, those who have died; question marks, unknown number; 0/1 indicates heterozygosity; 1/1 indicates homozygosity. Sanger sequencing representative chromatograms show that the nonsynonymous variant c.65C>G alters GCT (Ala) at that position to GGT (Gly) (**B**). The schematic diagram depicts the 3D structure of the human MYPN protein (Accession# Q86TC9) with a focus on the variant position (Ala22 in red font). It showcases the surrounding interactions (**C**) and highlights the specific location of each amino acid within that position. The green dashed lines indicate the hydrogen bonds between the included residues. Ala, Alanine; Gly, Glycine; Arg, Arginine; Ser, Serine; His, Histidine; Glu, Glutamate; Thr, Threonine; Tyr, Tyrosine; Leu, Leucine; H bond, Hydrogen bond.

**Figure 4 ijms-24-09119-f004:**
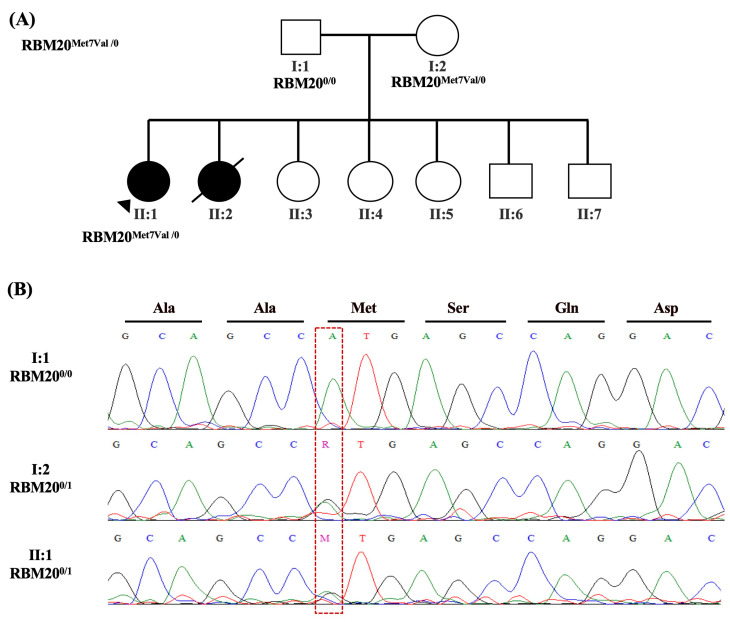
Patient-5 family pedigree and genetic analysis results. Family pedigree showing the family history of Patient-5 (II:1) and sequencing results in available family members for a rare heterozygous variant in *RBM20* (NM_001134363.3):c.19A>G (p.Met7Val) (**A**). Circles indicate female family members; squares, male family members; filled symbols, presence of the cardiomyopathy phenotype; open symbols, asymptomatic family members; arrows, the proband; symbols with a slash mark, those who have died; 0/1 indicates heterozygosity; 1/1 indicates homozygosity. Sanger sequencing representative chromatograms show that the nonsynonymous variant c.19A>G alters ATG (Met) at that position to GTG (Val). This substitution was transmitted to the proband from her mother (I:2) while the father carried the wild-type genotype (I:1) (**B**). The corresponding encoded amino acid is shown above the chromatograms, and the mutated nucleotide is enclosed in a red dashed box. Met, Methionine, Val, Valine; Ala, Alanine; Ser, Serine; Gln, Glutamine; Arg, Arginine; Asp, Aspartate.

**Table 1 ijms-24-09119-t001:** Patients’ clinical characteristics and genetic results.

ID	Cardiac Phenotype	Genetic Mutation	ACMG Classification Criteria	ACMG Classification Criteria Description	ACMG Classification-on	* MAF	^§^ Homo count	dbSNP rsID	Segregation Analysis Results	**Gene OMIM ID: Associated Cardiovascular-Related Disease**
**1**	Dilated cardiomyopathy	*NEXN* (NM_144573.4): c.1582_1584del (p. Glu528del)	PP3, BP3	*NEXN*: Gerp++ predicts conserved at this location, PhyloP predicts conserved at this location, *NEXN*: Transcript has inframe_deletion ontology, and is repeated 3 times	VUS/ Conflicting	1.5 × 10^4^	0	rs764505909	Homozygous in patient, and her sister.	613121: Cardiomyopathy
		*ABCC6* (NM_001171.6): c.3491G>A (p. Arg1164Gln)	PM2, PS1	*ABCC6*: Maximum allele frequency 0.009799% is below rare threshold of 0.01% for max AF source Annotate-gnomAD Exomes Variant Frequencies 2.1.1, BROAD, *ABCC6*: The variant causes the same pathogenic mutation as a pathogenic variant in the same amino acid for the following sources: ClinVar.	Likely pathogenic	2.39 × 10^5^	0	rs63750457	Heterozygous in patient, and her sister	603234: Atrial calcification and premature atherosclerosis; Pseudoxanthoma elasticum
		*MYH11* (NM_001040113.2): c.2383G>A (p. Asp795Asn)	PM2, PP2, PP3	*MYH11*: Maximum allele frequency 0.000879% is below rare threshold of 0.01% for max AF source Annotate-gnomAD Exomes Variant Frequencies 2.1.1, BROAD, *MYH11*: Missense variant in gene with disease commonly caused by missense variants, *MYH11*: Gerp++ predicts conserved at this location, PhyloP predicts conserved at this location, sift predicts this variant is: ’Damaging’, Polyphon predicts that this variant is ’Damaging’	VUS/Weak pathogenic	3.98 × 10^6^	0	rs768858261	Heterozygous in patient, and her sister	160745: Familial Thoracic Aortic Aneurysm and Aortic Dissection
		*TCAP* (NM_003673.4): c.113G>T (p. Cys38Phe)	PM2, BP1	*TCAP*: gnomAD genomes homozygous allele count = 0 is less than 2 for AD/AR gene *TCAP*, good gnomAD genomes coverage = 31.0. gnomAD exomes homozygous allele count = 0 is less than 2 for AD/AR gene *TCAP*, good gnomAD exomes coverage = 41.0, *TCAP*: 8 out of 22 non-VUS missense variants in gene *TCAP* are benign = 36.4% which is more than threshold of 33.1%	VUS/ Conflicting	1.08 × 10^4^	0	rs375310569	Heterozygous in patient, and her sister	604488: Cardiomyopathy
		*KCNJ11* (NM_000525.4): c.112A>G (p. Lys38Glu)	PM2, PP2, PM1, PP3	*KCNJ11*:Variant is missing from all sub population sources.,*KCNJ11*:Missense variant in gene with disease commonly caused by missense variants, *KCNJ11*:There are no benign variants within 3 amino acids of this variant, and 2 pathogenic/0 benign variants within 6 amino acids of this variant.,*KCNJ11*:Gerp++ predicts conserved at this location, PhyloP predicts conserved at this location, Sift predicts this variant is: ’Damaging’, Polyphon predicts that this variant is ’Damaging’	Likely pathogenic	0	0		Heterozygous in patient, and her sister	600937: Sudden Cardiac Death
**2**	Dilated cardiomyopathy	*NEXN* (NM_144573.4): c.1582_1584del (p. Glu528del)	PP3, BP3	NEXN: Gerp++ predicts conserved at this location, PhyloP predicts conserved at this location, *NEXN*: Transcript has inframe_deletion ontology, and is repeated 3 times	VUS/ Conflicting	1.53 × 10^4^	0	rs764505909	Homozygous in patient, and her sister. Heterozygous in all patient’s children	613121: Cardiomyopathy
		*ABCC6* (NM_001171.6): c.3491G>A (p.Arg1164Gln)	PM2, PS1	*ABCC6*: Maximum allele frequency 0.009799% is below rare threshold of 0.01% for max AF source Annotate-gnomAD Exomes Variant Frequencies 2.1.1, BROAD, *ABCC6*: The variant causes the same pathogenic mutation as a pathogenic variant in the same amino acid for the following sources: ClinVar.	Likely pathogenic	2.39 × 10^5^	0	rs63750457	Heterozygous in patient, her sister and her 4 children	603234: Atrial calcification and premature atherosclerosis; Pseudoxanthoma elasticum
		*MYH11* (NM_001040113.2): c.2383G>A (p. Asp795Asn)	PM2, PP2, PP3	*MYH11*: Maximum allele frequency 0.000879% is below rare threshold of 0.01% for max AF source Annotate-gnomAD Exomes Variant Frequencies 2.1.1, BROAD, *MYH11*: Missense variant in gene with disease commonly caused by missense variants, *MYH11*: Gerp++ predicts conserved at this location, PhyloP predicts conserved at this location, sift predicts this variant is: ’Damaging’, Pollyphen predicts that this variant is ’Damaging’	VUS/Weak pathogenic	3.98 × 10^6^	0	rs768858261	Heterozygous in patient, her sister and her 4 children	160745: Familial Thoracic Aortic Aneurysm and Aortic Dissection
		*TCAP* (NM_003673.4): c.113G>T (p. Cys38Phe)	PM2, BP1	*TCAP*: gnomAD genomes homozygous allele count = 0 is less than 2 for AD/AR gene *TCAP*, good gnomAD genomes coverage = 31.0. gnomAD exomes homozygous allele count = 0 is less than 2 for AD/AR gene *TCAP*, good gnomAD exomes coverage = 41.0, *TCAP*: 8 out of 22 non-VUS missense variants in gene *TCAP* are benign = 36.4% which is more than threshold of 33.1%	VUS/ Conflicting	1.08 × 10^4^	0	rs375310569	Heterozygous in patient, her sister and her 3 children	604488: Cardiomyopathy
		*KCNJ11* (NM_000525.4): c.112A>G (p. Lys38Glu)	PM2, PP2, PM1, PP3	*KCNJ11*:Variant is missing from all sub population sources.,*KCNJ11*:Missense variant in gene with disease commonly caused by missense variants,*KCNJ11*:There are no benign variants within 3 amino acids of this variant, and 2 pathogenic/0 benign variants within 6 amino acids of this variant.,*KCNJ11*:Gerp++ predicts conserved at this location,PhyloP predicts conserved at this location, Sift predicts this variant is: ’Damaging’,Pollyphen predicts that this variant is ’Damaging’	Likely pathogenic	0	0		Heterozygous in patient, her sister and her 2 children	600937: Sudden Cardiac Death
**3**	Hypertrophic obstructive cardiomyopathy	*MYH7* (NM_000257.4): c.4066G>A (p. Glu1356Lys)	PM2, PP2, PM1, PS1, PP3	*MYH7*:Variant is missing from all sub population sources., *MYH7*:Missense variant in gene with disease commonly caused by missense variants,*MYH7*:There are no benign variants within 3 amino acids of this variant, and 2 pathogenic/0 benign variants within 6 amino acids of this variant.,*MYH7*:The variant causes the same pathogenic mutation as a pathogenic variant in the same amino acid for the following sources: ClinVar.,*MYH7*:Gerp++ predicts conserved at this location,PhyloP predicts conserved at this location, Sift predicts this variant is: ’Damaging’,Pollyphen predicts that this variant is ’Damaging.	Pathogenic	0	0	rs727503246|	Heterozygous in patient and her son.	160760: Cardiomyopathy
**4**	Dilated cardiomyopathy	*MYPN* (NM_001256267.1): c.65C>G (p. Ala22Gly)	BP4, BP1, PM2	*MYPN*: MetaRNN = 0.034 is between 0.00692 and 0.108 ⇒ strong benign. *MYPN*:92 out of 122 non-VUS missense variants in gene *MYPN* are benign = 75.4% which is more than threshold of 33.1%. *MYPN*: gnomAD genomes homozygous allele count = 0 is less than 2 for AD/AR gene *MYPN*, good gnomAD genomes coverage = 30.9. gnomAD exomes homozygous allele count = 0 is less than 2 for AD/AR gene *MYPN*, good gnomAD exomes coverage = 72.5.	Likely benign	1.67 × 10^4^	0	rs145142157	Heterozygous in patient, her parents, and several siblings	608517: Cardiomyopathy
**5**	Dilated cardiomyopathy	*RBM20* (NM_001134363.3): c.19A>G (p. Met7Val)	PM2, PP2	*RBM20*: Variant is missing from all sub population sources., *RBM20*: Missense variant in gene with disease commonly caused by missense variants	VUS/Likely Benign	0	0		Heterozygous in patient and her mother	613171: Cardiomyopathy
		*RBM20* (NM_001134363.3): c.1881-3C>T	BA1, BS2, PP3, BP7, BP6	*RBM20*: Allele frequency is above 0.005 dominant threshold for source Annotate-gnomAD Exomes Variant Frequencies 2.1.1, BROAD., *RBM20*: Allele counts greater than 0 for dominant gene., *RBM20*: Gerp++ predicts conserved at this location, PhyloP predicts conserved at this location, *RBM20*:0 of 4 algorithms predict a disrupted splice site, *RBM20*: ClinVar Annotation has reputable ’Likely Benign’ classification	Likely benign	9.98 × 10^4^	0	rs138436392	Heterozygous in patient and her father	
		*FLNC* (NM_001458.5): c.3004C>T(p. Arg1002Trp)	PP2, PP3	*FLNC*: Missense variant in gene with disease commonly caused by missense variants, *FLNC*: Gerp++ predicts conserved at this location, PhyloP predicts conserved at this location, sift predicts this variant is: ’Damaging’, Pollyphen predicts that this variant is ’Damaging’	VUS/Weak Pathogenic	2.64 × 10^5^	0	rs555764780	Heterozygous in patient and her mother	102565: Cardiomyopathy

* MAF: Minor Allele frequency in GnomAD exomes; ^§^ Homo count: Homozygous allele count in GnomAD exomes.

## Data Availability

Data is unavailable due to privacy restrictions.

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
