# Peer review of "Introducing and Implementing Genetic Assessment in Cardio-Obstetrics Clinical Practice: Clinical and Genetic Workup of Patients with Cardiomyopathy"

_ijms, 2023, doi:10.3390/ijms24119119_

Round 1
Reviewer 1 Report
The article is well-written and interesting. It's easy to read and scientifically and clinically attractive.
Author Response
The article is well-written and interesting. It's easy to read and scientifically and clinically attractive.
We thank Reviewer 1 for this positive comment.
Reviewer 2 Report
This article is devoted to an interesting study based on the analysis of a series of clinical observations in cardiovascular pathology in pregnant women. The article is written according to the classical principle. Observation representations are of interest to researchers. The article is made at a high scientific and technical level. There are some comments. The methods are described quite fully.
Maybe expand the discussion a little, cite more sources of literature, and find common unifying points. It can be analyzed by the type of cardiomyopathy and the possible relationship between the course of pregnancy, weakness of labor, obstetric anamnesis.
Considering that there are researchers from the UK among the co-authors, the English language is good, there is no doubt
Author Response
Reviewer 2
This article is devoted to an interesting study based on the analysis of a series of clinical observations in cardiovascular pathology in pregnant women. The article is written according to the classical principle. Observation representations are of interest to researchers. The article is made at a high scientific and technical level. There are some comments. The methods are described quite fully.
Maybe expand the discussion a little, cite more sources of literature, and find common unifying points. It can be analyzed by the type of cardiomyopathy and the possible relationship between the course of pregnancy, weakness of labor, obstetric anamnesis.
We thank Reviewer 2 for this valuable suggestion. We modified the Discussion Section significantly. We expanded the discussion, cited more references, and unified the discussed points under subtitles.
Reviewer 3 Report
In the manuscript 'Introducing and Implementing Genetic Assessment in Cardio-Obstetrics Clinical Practice: Clinical and Genetic Workup of Patients with Cardiomyopathy', submitted by Mansoori and coworkers the authors present interesting families carrying some variants in cardiomyopathy genes like NEXN or MYH7. Unfortunately, the authors do not present any functional or structural data and therefore, the manuscript is completely descriptive and not really sufficient for publication in IJMS. Here some points which should support my suggestion to reject this manuscript:
1.) The authors should follow strictly the guidelines of the ACMG to classify the mutations.
2.) The MAFs of the variants should be indicated and discussed.
3.) What is with functional or structural tests to support the classification of these variants?
4.) Mutation in ABCC6 cause PXE... A fact which should be discussed by the authors. What is with the impact of these variants? Are they known from the literature? Have these variants modifying effects?
5.) The authors should discuss the genetic background of non-ischemic cardiomyopathies including the most relevant genes ... See for example for an overview the following book chapter: Gerull, B., Klaassen, S., & Brodehl, A. (2019). The genetic landscape of cardiomyopathies. Genetic Causes of Cardiac Disease, 45-91.
6.) All gene names should be written in Italics.
7.) Please indicate in the electropherograms within the figures also the coding codons and the encoded amino acids.
8.) Table 1: RBM20 not BRM20.
9.) Is there any splicing defect caused by the described RBM20 variants?
10.) How robust are the Alphafold models? Is the structure affected by the mutations?
11.) Are the NEXN mutations affection contractility or actin-polymerization? Again, no functional data were presented. Could the affects be modelled using animal models or cell culture models?
In summary, the cases are interesting but deserve a lot of more functional work before publication. I suggest that the authors analyse the interesting variants in more detail and resubmit then a revised manuscript. Because there are absolutely no functional data presented, I think that a rejection at this point is necessary.
The English needs some moderate changes.
Author Response
Reviewer 3
In the manuscript 'Introducing and Implementing Genetic Assessment in Cardio-Obstetrics Clinical Practice: Clinical and Genetic Workup of Patients with Cardiomyopathy', submitted by Mansoori and coworkers the authors present interesting families carrying some variants in cardiomyopathy genes like NEXN or MYH7. Unfortunately, the authors do not present any functional or structural data and therefore, the manuscript is completely descriptive and not really sufficient for publication in IJMS. Here some points which should support my suggestion to reject this manuscript:
Dear Respected Reviewer,
We appreciate your insightful comments and suggestions regarding our manuscript. We acknowledge your request for functional studies on the detected variants, as they are crucial for understanding the biological effects of genetic variants. However, obtaining samples for functional studies without causing harm to the patients is challenging, as the genes corresponding to the detected variants are mainly functional in the heart.
While we understand the importance of animal modeling and functional experiments, we currently lack the necessary expertise and resources to perform these experiments. Nevertheless, we utilized in silico tools to predict the potential functional effects of the detected variants, which we have described in detail in the manuscript.
Our study addresses the critical gap in knowledge about the prevalence and spectrum of cardiac conditions in pregnant women in the United Arab Emirates. We emphasize the importance of a multidisciplinary approach to diagnosis and treatment. To illustrate the significance of such an approach, we comprehensively analyze five women's cases from our retrospective study. Our findings highlight the pivotal role of collaborative teamwork in minimizing adverse cardiac outcomes and reducing mortality rates during pregnancy.
We hope our below point-by-point responses address your concerns and highlight the significance of our study. Thank you again for your valuable comments and suggestions, and we look forward to hearing from you soon.
- The authors should follow strictly the guidelines of the ACMG to classify the mutations.
We added the ACMG exact classification, classification criteria, and criteria description for each variant in Table1_1 as computed by VarSeq Suite ACMG Classifier algorithm.
- The MAFs of the variants should be indicated and discussed.
We added minor allele frequencies (MAFs) to all mutations listed in Table 1 and in the text of the main Manuscript as estimated by gnomAD database.
- What is with functional or structural tests to support the classification of these variants?
Although we acknowledge the crucial role of animal modeling and functional experiments in validating our research findings, we regret to inform you that we lack the necessary expertise and resources to conduct such experiments. Therefore, we are unable to comply with your request at this time. However, we have already highlighted the limitations of our study in the Discussion Section (Strengths and limitations), where we have provided additional details on the detected variants and genes. We have also elaborated on these findings in the Results and Discussion Sections to offer a comprehensive overview of our research. We also modified the Abstract to reflect the focus of the study.
- Mutation in ABCC6 cause PXE... A fact which should be discussed by the authors. What is with the impact of these variants? Are they known from the literature? Have these variants modifying effects?
In the Discussion Section and Table 1 of the revised manuscript, we thoroughly examined this variant and its potential role in developing cardiomyopathy in our patients. Previous studies have reported this mutation in PXE cases, either in a homozygous or compound heterozygous state, with another ABCC6 variant not detected in our patients. However, we found no clinical evidence to suggest that PXE is a probable diagnosis for our patients. Nevertheless, we have informed the patients about this finding and advised them to seek consultation with a rheumatologist to address any potential issues that may arise.
- The authors should discuss the genetic background of non-ischemic cardiomyopathies including the most relevant genes ... See for example for an overview the following book chapter: Gerull, B., Klaassen, S., & Brodehl, A. (2019). The genetic landscape of cardiomyopathies. Genetic Causes of Cardiac Disease, 45-91.
We discussed the genetic background of cardiomyopathies in the Discussion Section and cited this book chapter.
- All gene names should be written in Italics.
We corrected all gene names to Italics.
- Please indicate in the electropherograms within the figures also the coding codons and the encoded amino acids.
We modified all Figures. We indicated all encoded amino acids and the site of mutations on the chromatograms.
- Table 1: RBM20 not BRM20.
We apologize for this typo. We corrected this mistake in Table 1 and through the Main Manuscript.
- Is there any splicing defect caused by the described RBM20 variants?
At first, we believed it was necessary to mention this nucleotide due to its conservation as determined by Gerp++, its location in a splice region (-3), and its representation as the second hit in the RBM20 gene. However, to address the Reviewer's concerns, we conducted a thorough bioinformatics analysis using four commonly used splice site detection tools: NNSplice, MaxEntScan, GeneSplicer, and HumanSplicingFinder. Surprisingly, none of these algorithms predicted any splicing disruption caused by this rare variant. Consequently, we have removed it from the main manuscript and instead included it solely in Table 1 as a detected variant in the patient.
- How robust are the Alphafold models? Is the structure affected by the mutations?
To better showcase the possible effects of the observed changes, we have provided zoomed-in visualizations of the 3D structures of the relevant proteins in UniProtKB. These models allow us to examine the surrounding interactions that may be impacted by the observed alteration and provide a more detailed understanding of the potential consequences of the variant.
- Are the NEXN mutations affection contractility or actin-polymerization? Again, no functional data were presented. Could the affects be modelled using animal models or cell culture models?
It is uncertain whether the observed mutation impacts NEXN contractility or actin polymerization. We regret to inform you that we lack the necessary resources and expertise to model this variant in animals or cells. However, we have included literature references to support the significance of this variant, which has been reported in two cases of cardiomyopathy in a homozygous state, as in our two patients.
Round 2
Reviewer 3 Report
Some points were corrected by the authors. However, functional data about the identified mutations are still not present in this manuscript.